## Article

# Uniform Illumination Using Single-Surface Lens through Wavefront Engineering

Aria Moaven [1] , Hamid Pahlevaninezhad [2,3], Masoud Pahlevaninezhad [1] and Majid Pahlevani [1,*]

1 Department of Electrical and Computer Engineering, Queen's University, Kingston, ON K7L 3N6, Canada
2 Massachusetts General Hospital, Harvard Medical School, Boston, MA 02114, USA
3 Harvard John A. Paulson School of Engineering and Applied Sciences, Harvard University, Cambridge, MA 02134, USA
* Correspondence: majid.pahlevani@queensu.ca

**Abstract:** Recent advancements in high power light-emitting diode (LED) technology have presented greenhouse industry with a more efficient and flexible alternative light source for horticulture. However, the light distribution on the plant remains a challenge that has notable implications on the plant growth. A non-uniform light distribution on the canopy with hot spots is well-known to adversely affect the yield. Here, we present a technique to engineer the light wavefront of a solid-state source using a single-surface optics, which yields a highly uniform light distribution across the plant. This technique achieves over 90% illuminance uniformity, preserved at various distances from the source, for a cone of light with an up to 120° angular range. This work aids the paradigm shift towards LEDs as a competitive light source in horticulture.

**Keywords:** horticulture lighting; LED grow-light; LED non-uniformity; uniform illumination; freeform lens





## 1. Introduction

Haitz law [1], the analog of Moore's law in the semiconductor industry, anticipates a twenty-fold increase in the output power and a ten-fold decline in the cost of LEDs every decade. This projection indicates that LEDs are poised to dominate the lighting industry as is already the case in several applications such as indoor lighting, street lighting, and display technology.

The greenhouse industry sorely requires a more efficient and more flexible alternative to high pressure sodium (HPS) light sources which could be met with high-power LEDs [2–8]. However, achieving an equivalent light intensity on plant canopy necessitates a large number of LEDs, yielding an LED grow-light panel economically not competitive with HPS sources [4,9]. Techniques for more efficient use of optical energy emitted by LEDs would certainly mitigate this challenge. In greenhouses, non-uniform lighting of LEDs results in either inadequate or excessive lighting at certain locations [10]. Increasing light intensity can boost plants' growth and reproduction, however, excessive light level leads to growth saturation after a threshold point [11,12]. This issue gives rise to a suboptimal design of optical output power to ensure sufficient illumination throughout the plant canopies [12–15].

Freeform optics often uses one or more refractive surfaces to redirect the emitted light from a source to any desired illumination on the target plane [16–18]. The ray-mapping method involves calculating a mapping between the source and the target light distributions and constructing the freeform surface as a result [19]. Ray-mapping is the most straightforward and, in turn, the most common technique for designing freeform optics amongst others [20]. However, there is no implicit constrain in ray-mapping to enforce an integrable freeform surface solution, a condition required for the surface to be practically realizable [19]. Even if ray-mapping solutions yield an integrable solution, they

can only achieve uniform illumination for a narrow light cone due to notable numerical errors in surface construction [16,19,21–32].

In this work, we present an alternative ray-mapping algorithm to design single surface freeform lenses. This technique yields highly uniform illumination using a physically realizable surface by enforcing integrability. In addition, eliminating construction errors makes this technique applicable to the sources with wide angular emission. Here, we experimentally report 90.4% uniformity for a 120° light cone.

## 2. Materials and Methods

Similar to all ray-mapping methods, this work relies on the energy conservation principle imposing the preservation of optical power emitted from a fixed light cone on the target plane [30,33]. Ray-mapping techniques determine a diffeomorphism of the source light distribution to a desired light distribution on the target plane (an imaginary far-field plane). A suitable ray-mapping needs to be integrable and outputs a smooth lens surface specially for non-paraxial or off-axis beams as non-smooth free-form lenses are either impossible or costly to fabricate [18,19,22,33]. Here, without loss of generality, we present a specific algorithm to transform a Lambertian point source (e.g., LED) irradiance into a circular uniform distribution.

A ray-mapping method generally entails two main steps: (1) the definition of a mapping, which associates each incident ray with a particular outgoing ray, and (2) the determination of the refracting surface, which yields such ray-mapping.

(1) To find the mapping, we assume a hypothetical plane (principal plane) at which all rays are bent to yield the desired illumination pattern on the plant canopy (target plane). The LED's irradiance on the principal plane is proportional to the inverse fourth power of the distance each ray travels from the LED to the principal plane (since the LED emits a Lambertian beam perpendicular to the principal plane). Our technique divides the principal plane into N rings, each receiving an equal amount of light power. Due to the lower irradiance that the outer rings receive, their widths are increasingly wider (Figure 1). To make a uniform illumination on the target plane, the rays impinging on the principal plane need to be bent such that they strike the target plane on equally-thick rings (L). Crucially, the rings' sequence must be preserved on both planes to impose integrability.

(2) Given the incident and outgoing rays' positions and angles, the corresponding refractive surfaces can be determined at the principal plane. As an intermediate step, the required change in the lateral optical momentum at the principal plane is calculated. This change in lateral momentum alters the incident wavefront (red solid curve in Figure 2a) into the refracted wavefront (blue solid curve in Figure 2a), which differs from that otherwise resulted from free-space propagation (dotted red curve in Figure 2a). This procedure is known as wavefront tailoring. Explicitly, the optical path differences (OPDs) between the rays traversing from the incident wavefront to the refracted wavefront are calculated (purple curve in Figure 2b); then, the corresponding lateral momentum change required at the principal plane is determined (red curve in Figure 2b); finally, the refractive surface is found using Snell's law.

Using the equation below, each ray's free-space OPD can be calculated:

$$\text{OPD} = L_{\text{chief}} - L_{\text{sp}} - L_{\text{pt}}, \tag{1}$$

where $L_{\text{chief}}$, $L_{\text{sp}}$ and $L_{\text{pt}}$ correspond to chief ray (the shortest), source to principal plane, and principal plane to target optical path lengths, respectively. Figure 2a demonstrates the OPDs of some rays (purple lines).

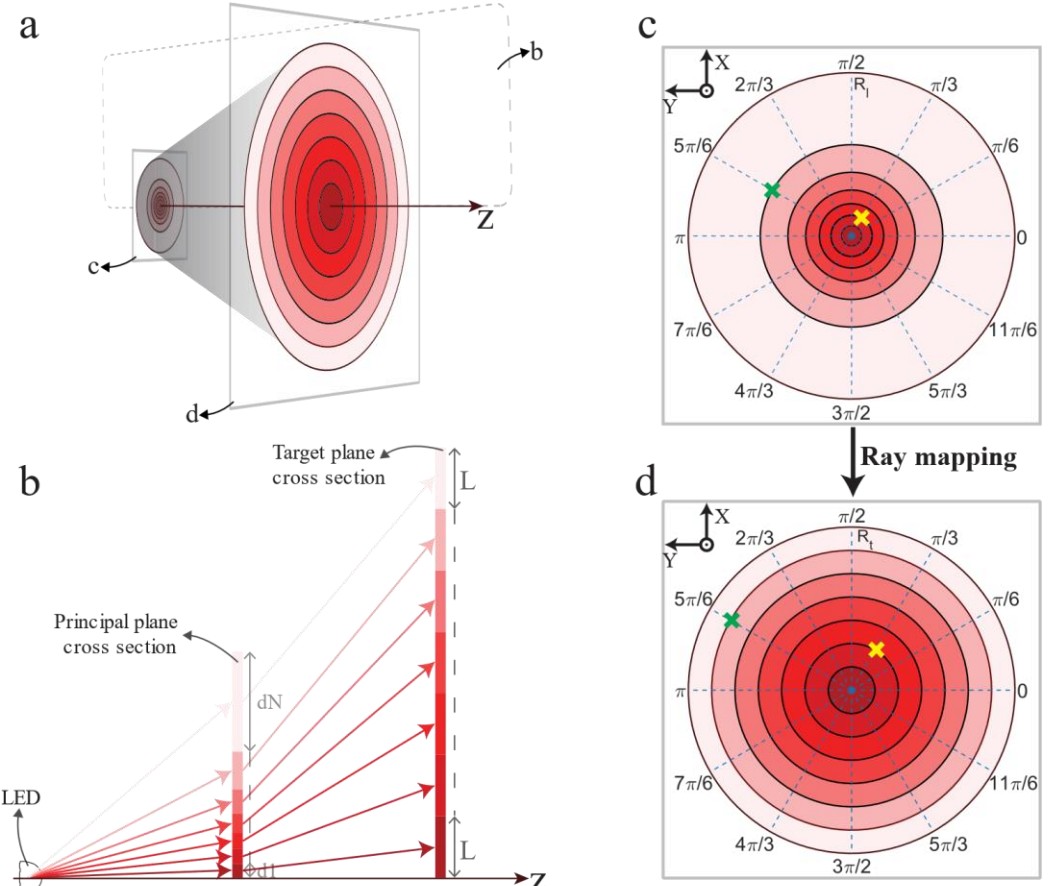

**Figure 1.** Ray-mapping algorithm demonstration. (**a**) 3D presentation of the principal plane and target plane. (**b**) Cross section of the LED, the principal plane, and the target plane. d1 to dN are the widths of each ring on the principal plane, while L is the width of the corresponding rings on the target plane. (**c**) Principal plane rings and sample points (yellow and green crosses). (**d**) Corresponding mapped rings and sample points on the target plane.

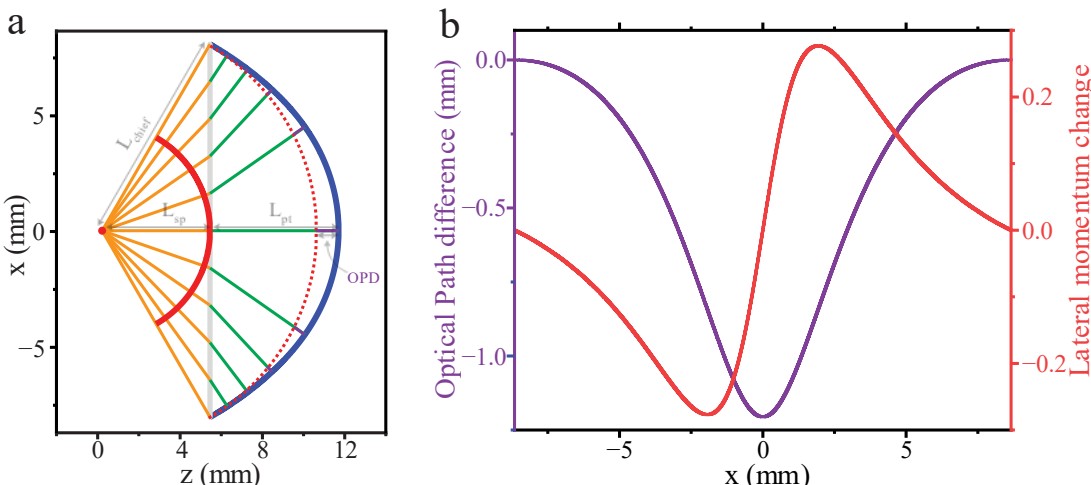

**Figure 2.** Wavefront tailoring process. (**a**) cross section of the incident (solid red curve), refracted (blue curve) and free-space propagated (dashed red curve) wavefronts to calculate optical path differences. Lens to principal plane and principal plane target plane rays have been illustrated with orange and green line, respectively. (**b**) Required OPD based on wavefront tailoring (purple curve) for each point of the principal plane cross section (gray line on (**a**)) and the corresponding lateral momentum change on the lens cross section (red graph).

The lateral momentum change ($\Delta p$) corresponds to the required bending of each ray on the principal plane's cross-section, which can be calculated using the equation as follows:

$$\Delta p = \frac{\partial(\text{OPD})}{\partial x} \qquad (2)$$

The Figure 2b illustrates the OPD (purple curve) and its corresponding change in lateral momentum (red curve) along the x-axis on the principal plane.

The point-by-point calculation of lens surface given the bending angles is illustrated in Figure 3a. The construction of the lens surface begins from the initial point corresponding to the marginal ray (Figure 2a) which coincides with the principal plane that corresponds to the marginal ray (Figure 2a). The consecutive points are shifted along their incident ray such that the lens surface remains continuous (Figure 3a). The entire lens surface can be determined iteratively using this paradigm. The inner surface of the lens is a hemisphere to preserve the angle of incident rays from the source. This ray-mapping method, which satisfies the integrability yields a smooth manufacturable surface.

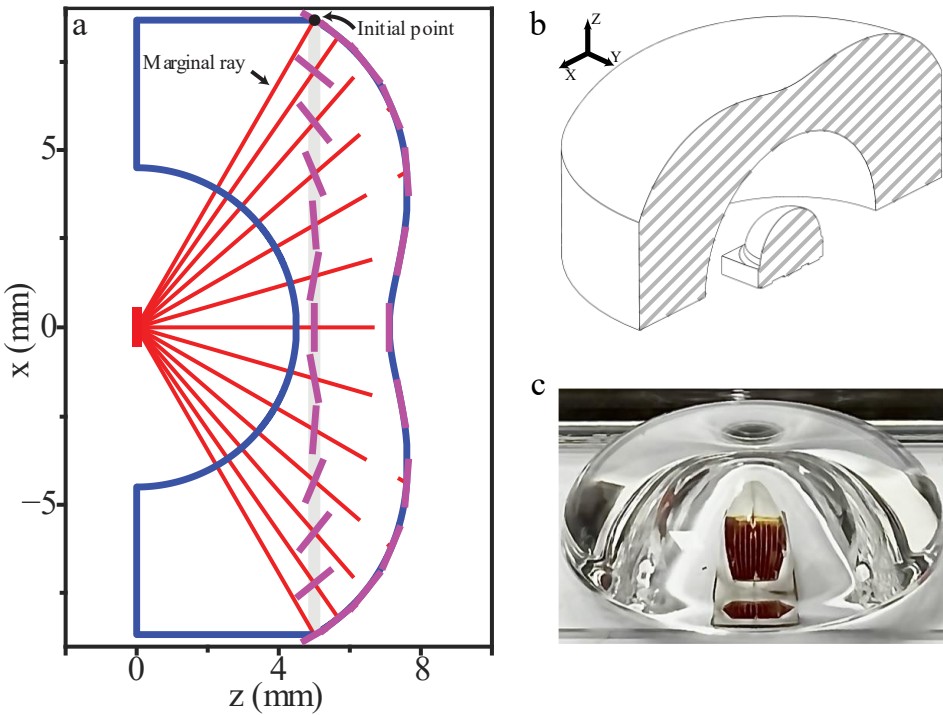

**Figure 3.** Lens surface construction. (**a**) 2D presentation of the numerical lens surface calculation by translation of refraction surface (purple lines) on principal plane (gray line) to the lens surface. (**b**) Final lens design cross-section on xz-plane. (**c**) manufactured lens and the LED assembly image.

The lens material selected for this study was poly(methyl methacrylate) (PMMA), due to its excellent optical properties including low absorption and relatively consistent refractive index in the visible wavelength range. In addition, PMMA has desirable mechanical properties such as high temperature resistance, low thermal expansion, and low humidity absorption making it suitable for greenhouse lighting applications. The prototype lens was manufactured using CNC milling and polishing machine (OPS INGERSOLL (Burbach, Germany) EAGLE V9 5-AXIS), resulting in an arithmetic average roughness (Ra) of 20–30 nm.

The designed lens was computationally verified using ZEMAX (Kirkland, WA, USA) OpticStudio software, considering a specific LED emission profile (OSRAM (Premstätten, Austria) OSLON® Square, GH CSSRM4.24). In this simulation, the target plane is a square-shaped area of 1 m² located 25 cm from the source, perpendicular and centered to the

optical axis. The detector plane (target plane) resolution is 200 by 200, and the lens refractive index is assumed to be 1.4880 (PMMA @660 nm). Analyzing the results and calculating the uniformity and power in the specified target plane were carried out using MATLAB.

To experimentally demonstrate the lens performance, we utilized a setup comprised of a photo-detector (Thorlabs (Newton, MA, USA) PDA36A) mounted on a 2-dimensional translation stage capable of scanning the entire target plane (Figure 4). The setup continuously scans a 1 m-by-1 m square-shaped target plane using two stepper motors.

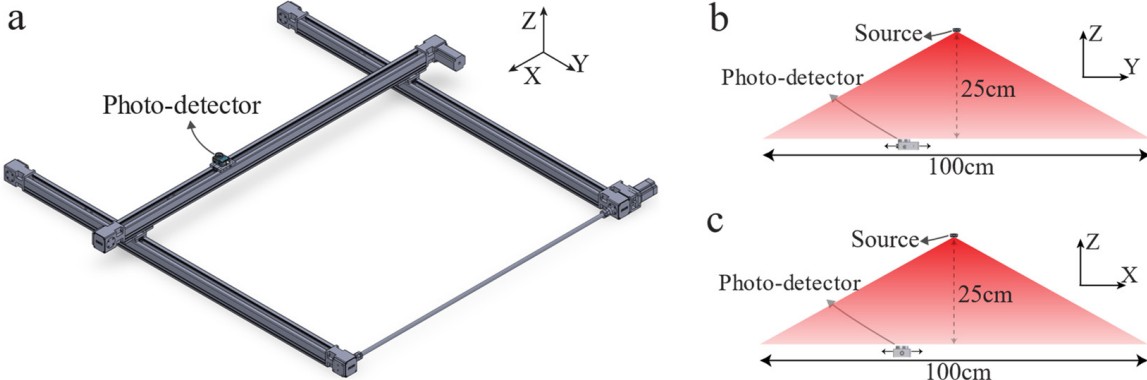

**Figure 4.** The experimental setup computer-aided design (CAD). (**a**) Dimetric view of the photodetector and the 2D translation stage, which measures the light intensity and its uniformity. (**b**) 2D view (yz-plane) of the setup. The source placed 25 cm above the center point of the scanner to obtain experimental results. (**c**) 2D view (xz-plane) of the setup. The source placed 25 cm above the center point of the scanner to obtain experimental results.

## 3. Results and Discussion

As the simulation results, Figure 5a,b compare the light intensities distribution on the target plane with and without the lens, respectively. Here, the light uniformity index is the ratio of the minimum to the maximum light intensity on the target plane. As illustrated in Figure 5c, the result indicates 95.1% light uniformity for a 120° light cone using the refractive lens, significantly higher uniformity than what can be achieved without the lens (11.2%). The power received at the target plane remains the same with and without the refracting surface, verifying the power efficiency of the designed lens.

Experimentally, the dimensions and the position of the target plane match those used in the simulation model. Figure 5d,e illustrate the light intensities on the target plane with and without the lens, respectively. The result indicates 90.6% uniformity, slightly lower than the computational expectation, possibly due to the minor misalignment in the setup, the manufacturing errors, and Fresnel losses. Expectedly, the optical power measured on the target plane was approximately 7% lower than the simulation result due to the absorption of the lens material.

The uniform illumination pattern of a single LED does not necessarily reflect the uniformity of an entire lamp consisting of an array of LEDs. However, due to the smaller distance between the LEDs (~1.5 cm) compared to the distance between the lamp and the target plane (more than 25 cm), the uniformity will remain above 90% for most of the target plane (Figure 6). The simulation and the experimental results for an array of 1 by 25 LEDs located 25 cm away from the target plane are shown in Figure 6a,c, respectively. Increasing distance from the target plane reduces the effect of an LED array's spatial extension on the target plane illumination uniformity (Figure 6b,d). Figures 5e and 6d (or Figures 5b and 6b) demonstrate similar illumination patterns, supporting our assertion about the light uniformity of a whole lamp when the target plane is far enough.

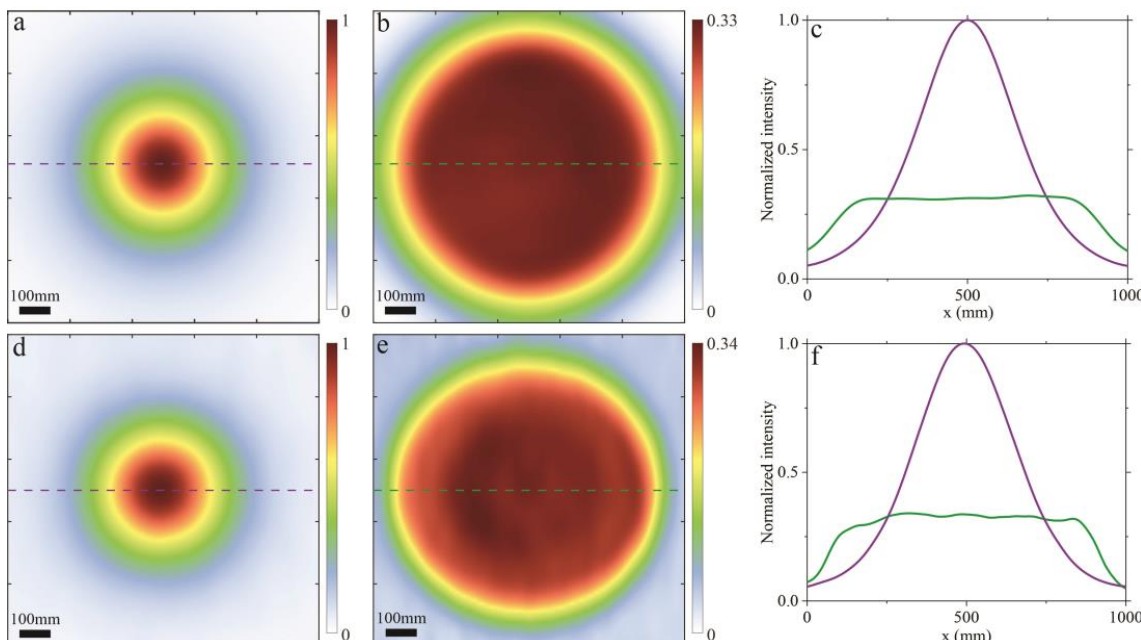

**Figure 5.** Simulation and experimental results for a target plane located 25 cm away from the source. (**a**) ZEMAX simulation result for OSRAM LED rayfile without the lens. Dashed purple line indicates the middle row. (**b**) Simulation result for OSRAM LED rayfile with the lens. (**c**) Comparing middle rows' irradiances (dashed horizontal lines on Figure 2a,b) of the simulation result with (green curve) and without (purple curve) the lens. (**d**) Normalized irradiance for experimental result without the lens. (**e**) Normalized irradiance for experimental result with the lens. (**f**) Comparing middle rows' irradiances of the experimental result with (green line) and without (purple line) the lens.

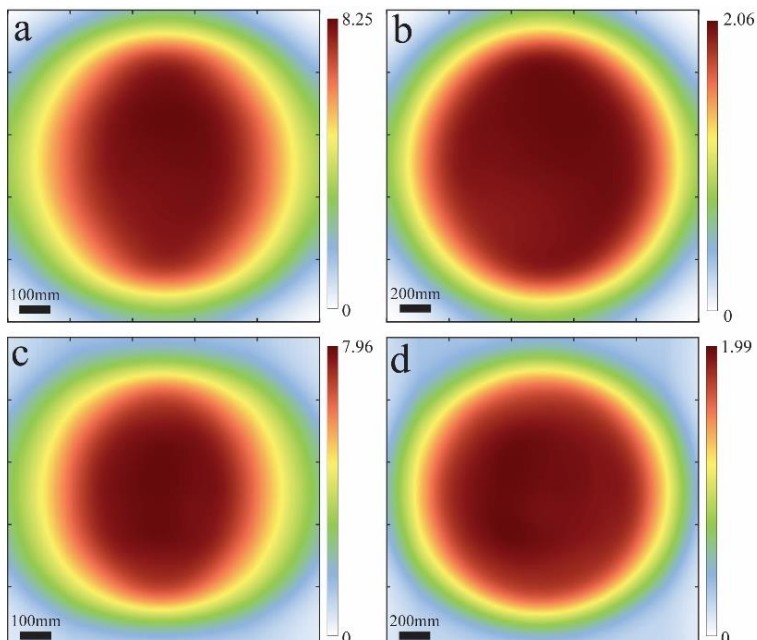

**Figure 6.** Simulation and experimental results for a 1 by 25 LED array, oriented along the x-axis. (**a**) Simulation result for a 1 m² target plane, located 25 cm away from the LED array. (**b**) Simulation result for a 4 m² target plane, located 50 cm away from the LED array. (**c**) Experimental result for a 1 m² target plane, located 25 cm away from the LED array. (**d**) Experimental result for a 4 m² target plane, located 50 cm away from the LED array.

In greenhouse lighting systems, a high light uniformity translates to the use of fewer LEDs (~3-times) to achieve the same intensity level at the plant periphery. Unlike the-state-of-the-art techniques [23,28,30,34–36], the method presented here can be manufactured with standard techniques (injection molding or CNC machining), works for light sources with a wide angular range and different colors, and has no constraint regarding the source-target plane distance.

## 4. Conclusions

The greenhouse industry demands more uniform lighting systems to maximize the plants' yield. This work introduces a simple algorithm to design a single refractive surface that uniformly distributes the light intensity emitted by wide angle commercial LEDs. This algorithm enforces the surface integrability, finds the lateral optical momentum change at the principal plane, and determines a refracting surface which realizes the desired mapping. This ray-mapping algorithm is general and can be utilized for designing illumination systems with different shapes and different lens types. Using this method increases of the light distribution uniformity from 11.2% to over 90%, enabling more efficient lighting and potentially higher growth yield.

## 5. Patents

A.M., H.P. and M.P. (Majid Pahlevani) are inventors on a relevant patent application (A8146800USP).

**Author Contributions:** Conceptualization, H.P., M.P. (Majid Pahlevani) and A.M.; methodology, A.M. and H.P.; software, A.M.; validation, A.M., H.P. and M.P. (Masoud Pahlevaninezhad); formal analysis, A.M.; investigation, A.M. and H.P.; resources, A.M. and H.P.; data curation, A.M.; writing—original draft preparation, A.M.; writing—review and editing, H.P., M.P. (Majid Pahlevani) and M.P. (Masoud Pahlevaninezhad); visualization, A.M.; supervision, M.P. (Majid Pahlevani) and H.P.; project administration, M.P. (Majid Pahlevani); funding acquisition, M.P. (Majid Pahlevani). All authors have read and agreed to the published version of the manuscript.

**Funding:** This research was funded by Natural Sciences and Engineering Research Council (NSERC) of Canada, grant number 392075 and MITACS, grant number IT17040.

**Data Availability Statement:** Upon a reasonable request, the corresponding author can provide you with the data supporting the plots in this article and other findings of the study.

**Acknowledgments:** M. Pahlevani Thanks Genoptic LED Inc. for providing financial support for this project.

**Conflicts of Interest:** The authors declare no conflict of interest.

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
