# Peer review of "Uniform Illumination Using Single-Surface Lens through Wavefront Engineering"

_horticulturae, doi:10.3390/horticulturae8111019_

Round 1

Reviewer 1 Report

Ladies and Gentlemen, your article addresses an important and difficult issue, which is the surface uniformity of lighting in a greenhouse. This uniformity is determined by hundreds/thousands of LED lamps, whose fluxes interfere with each other. One lamp is a few COB LEDs or dozens of medium-power LEDs. The attractiveness of a greenhouse LED lamp is not determined by the low cost of medium-power LEDs - only the parameters of the entire lamp, i.e. energy efficiency and flux uniformity. Most often, the flux of a greenhouse lamp is not emitted in a circle but in a rectangular field.  Therefore, it is easier to shape the flux with a rectangular cross-section when using many medium-power LEDs than a few high-power LEDs. LED greenhouse lamps, are lamps with several hundred watts of power, so it is very rare to use lamps with a single LED, sometimes with 4/6 COB LEDs. Thus, on the basis of testing a selected 3W LED, even with a very carefully designed spatial correction module, it is not possible to conclude about the uniformity of the entire lamp, much less the illumination of a greenhouse crop.

In your article as I do not understand why:
- the use of a large number of LEDs in a greenhouse lamp is uneconomical (line 34) - compare, for example, Philips solutions in this matter; - English keywords inaccurately reflect the essence of the article - referring to the issues of LED greenhouse lighting and its surface uniformity; - the 7% loss of the lens was obtained (line 136) and how this value was measured - in Figure 4 we see that the loss for the 500 mm center is at least 70% and the area under the green curve is more than 7% smaller than the area under the purple curve; - the final conclusions concern the lens design algorithm and do not refer to the results obtained in relation to the topic of the work; - on which basis the suitability of this algorithm for lens design was generalised, e.g. for COB LEDs or LED strips, which are used in LED greenhouse lamps.

Reviewer 2 Report

The manuscript entitled: “Uniform illumination using single-surface lens through wave-front engineering”

General comments:

In my point of view, this manuscript is highly relevant, since it describes the enormous potential for improving the efficacy of LEDs, to overthrow the use of power avid HPS. However, some issues should be addressed. In my opinion further information should be provided on how the simulation was performed? How its results were analyzed? What were the inputs for the model? Moreover, and more importantly, I was unable to understand how the authors manufactured the lens. What equipment was used? What were the operational conditions?

Now, I would really like to know what is the impact on a crop. How it improves plant yields in comparison to a conventional led, or even (more interesting), to a HPS as well? In my opinion this would considerably enhance the impact of this manuscript.

Minor Comments:

Line 32, is there any particular reason for “High Pressure Sodium” to be in capitals?

Line 54 to 62 does not possess any references to support the statements. Please include relevant and adequate literature.

Figure 1 is not referenced in the text prior to its appearance (it is only referenced on line 78), furthermore, several letters used in the figure are not defined in the figure caption, such as N and L. please include this information.

Line 112, PMMA acronym not adequately defined.

Line 123, CAD acronym not defined.

Figure 4 does not depict the light source position. Can the authors please include that in the figure, or provide additional information the text and figure caption of its position?

Line 126 to 132, this information, in my opinion should be present in the Materials and Methods section.

Reviewer 3 Report

This paper presents study on Uniform illumination using single-surface lens through wave-front engineering.

The topic of high power light-emitting diode (LED) technology have presented greenhouse industry with a more efficient and flexible alternative light source for horticulture is meaningful and relevant for an international audience. However, the manuscript missing information to present a clear scientific contribution and novelty is not clearly demonstrated.

Introduction: The ray-mapping algorithms to design single sur-face freeform lenses are briefly described and refered..

Methodology: The novel contribution is not clear or specifically defined. In some cases, the information is scarce or the calculation is not well defined. It is necessary to define in detail the process used in the construction method in detail: The wavefront tailoring process.

It would be appropriate to include the experimental setup image.

Line 66. The alternative ray-mapping algorithm proposed is not clearly defined.

Line 88. Lateral optical momentum is not defined, also it would be included in figure.

Line 108-109. It is necessary to clarify and define the "Wavefront tailoring process" and "Required optical path difference based on wavefront tailoring" in text. It is also necessary to include elements used in the figure.

Round 2

Reviewer 2 Report

The authors clearly improved the manuscript quality and scientific soundness.

Nevertheless, I must disagree with the authors, when they state in Response 2: "The effect of the illumination uniformity on the plant growth is certainly the most important result of this work. However, such verification entails meticulous studies including years of works and significant resources which makes it well beyond the scope of this manuscript". Arabidopsis thaliana, one of the most famous plant models has a life cycle of 12 weeks. Furthermore, if the authors intended to test a crop, such as Lactuca sativa, in a few weeks the authors would be able to see differences. I do really think this analysis is extremely important to enhance the manuscript impact.  Thus, in my opinion, the authors should make an effort to perform this analysis. If a collaboration is required to achieve it, please consider establishing one. Nevertheless, I leave the final decision to the authors and the Editor.

Author Response

As the reviewer correctly pointed out, the effect of uniform illumination on plant growth is a valuable study that improves the impact of this manuscript. However, there are some other studies that have examined the effect of the light intensity level on plant growth and reproduction. Plants dry out at different rates due to non-uniform lighting in a greenhouse. Managing crops that receive more light (and grow faster) with those that receive less light (and grow slower) becomes challenging in a large growing area [11]. According to one of these studies [10], increasing the light intensity leads to enhanced plant growth and reproduction, but after passing a threshold point, this enhancement becomes saturated. The proposed lens provides uniform light intensity along the plant canopies and this intensity could be adjusted to regulate the even growth of all plants in a greenhouse. To address this comment, some references have been added and the introduction section has been revised as follows:

In greenhouses, non-uniform lighting of LEDs results in either inadequate or excessive lighting at certain locations [10]. Increasing light intensity can boost plants' growth and reproduction, however, excessive light level leads to growth saturation after a threshold point [11,12]. This issue gives rise to a suboptimal design of optical output power to ensure sufficient illumination throughout the plant canopies [12-15].

Supporting references:

[10] RUNKLE, E. The Importance of Light Uniformity. 2017.

[11] Poorter, H.; Niinemets, Ü.; Ntagkas, N.; Siebenkäs, A.; Mäenpää, M.; Matsubara, S.; Pons, T. A meta‐analysis of plant responses to light intensity for 70 traits ranging from molecules to whole plant performance. New Phytologist 2019, 223, 1073-1105.

[12] Balázs, L.; Dombi, Z.; Csambalik, L.; Sipos, L. Characterizing the Spatial Uniformity of Light Intensity and Spectrum for Indoor Crop Production. Horticulturae 2022, 8, 644.

Reviewer 3 Report

Information has been added based on suggestions made made and the comments have been clarified

Round 3

Reviewer 2 Report

The manuscript entitled: Uniform illumination using single-surface lens through wavefront engineering, reference horticulturae-1894482

Is a relevant and now more honest manuscript. I do not have any opposition to its publication since the authors do not allude to plant growth rate that was not (unfortunately) tested.